# Night Photostimulation of Clearance of Beta-Amyloid from Mouse Brain: New Strategies in Preventing Alzheimer’s Disease

**DOI:** 10.3390/cells10123289

**Published:** 2021-11-24

**Authors:** Oxana Semyachkina-Glushkovskaya, Thomas Penzel, Inna Blokhina, Alexander Khorovodov, Ivan Fedosov, Tingting Yu, Georgy Karandin, Arina Evsukova, Dariya Elovenko, Viktoria Adushkina, Alexander Shirokov, Alexander Dubrovskii, Andrey Terskov, Nikita Navolokin, Maria Tzoy, Vasily Ageev, Ilana Agranovich, Valeria Telnova, Anna Tsven, Jürgen Kurths

**Affiliations:** 1Institute of Physics, Humboldt University, Newtonstrasse 15, 12489 Berlin, Germany; juergen.kurths@pik-potsdam.de; 2Department of Biology, Saratov State University, Astrakhanskaya 82, 410012 Saratov, Russia; thomas.pensel@charite.de (T.P.); inna-474@yandex.ru (I.B.); khorovodov2012@yandex.ru (A.K.); fedosov_optics@mail.ru (I.F.); engels2059@gmail.com (G.K.); arina-evsyukova@mail.ru (A.E.); eloveda@mail.ru (D.E.); adushkina.info@mail.ru (V.A.); shirokov_a@ibppm.ru (A.S.); paskalkamal@mail.ru (A.D.); terskow.andrey@gmail.com (A.T.); nik-navolokin@yandex.ru (N.N.); dethaos@bk.ru (M.T.); old-lon@yandex.com (V.A.); ilana.agranovich@yandex.ru (I.A.); ler.vinnick2012@yandex.ru (V.T.); anna.kuzmina.270599@mail.ru (A.T.); 3Sleep Medicine Center, Charité-Universitätsmedizin Berlin, Charitéplatz 1, 10117 Berlin, Germany; 4Britton Chance Center for Biomedical Photonics, Wuhan National Laboratory for Optoelectronics, Huazhong University of Science and Technology, Wuhan 430074, China; yutinting@hust.edu.cn; 5Collaborative Innovation Center for Biomedical Engineering, MoE Key Laboratory for Biomedical Photonics, School of Engineering Sciences, Huazhong University of Science and Technology, Wuhan 430074, China; 6Saratov Scientific Centre of the Russian Academy of Sciences (IBPPM RAS), Institute of Biochemistry and Physiology of Plants and Microorganisms, Prospekt Entuziastov 13, 410049 Saratov, Russia; 7Department of Pathological Anatomy, Saratov Medical State University, Kazachaya 112, 410012 Saratov, Russia; 8Department of Complexity Science, Potsdam Institute for Climate Impact Research, Telegrafenberg A31, 14473 Potsdam, Germany

**Keywords:** amyloid-β, night clearance, transcranial photostimulation, lymphatic system

## Abstract

The deposition of amyloid-β (Aβ) in the brain is a risk factor for Alzheimer’s disease (AD). Therefore, new strategies for the stimulation of Aβ clearance from the brain can be useful in preventing AD. Transcranial photostimulation (PS) is considered a promising method for AD therapy. In our previous studies, we clearly demonstrated the PS-mediated stimulation of lymphatic clearing functions, including Aβ removal from the brain. There is increasing evidence that sleep plays an important role in Aβ clearance. Here, we tested our hypothesis that PS at night can stimulate Aβ clearance from the brain more effectively than PS during the day. Our results on healthy mice show that Aβ clearance from the brain occurs faster at night than during wakefulness. The PS course at night improves memory and reduces Aβ accumulation in the brain of AD mice more effectively than the PS course during the day. Our results suggest that night PS is a more promising candidate as an effective method in preventing AD than daytime PS. These data are an important informative platform for the development of new noninvasive and nonpharmacological technologies for AD therapy as well as for preventing Aβ accumulation in the brain of people with disorder of Aβ metabolism, sleep deficit, elderly age, and jet lag.

## 1. Introduction

Amyloid-β (Aβ) is present in the brain’s interstitial fluid (ISF) and is considered a metabolic waste product [1]. The extracellular deposition of Aβ aggregates, the main constituent of senile plaques, is considered to be a pathological hallmark of Alzheimer’s disease (AD), which contributes to neuronal dysfunction and behavioral changes [2,3]. The mechanisms by which Aβ is cleared from the brain are not completely understood [4], although there is evidence that sleep plays an important role in Aβ clearance [5]. In rodents, chronic sleep restriction increases the Aβ levels in the ISF [6]. In healthy humans, positron emission tomography with 18F-florbetaben reveals that only one night of sleep deprivation is associated with an increase in the Aβ brain level by 5% [7]. Other imaging studies on healthy volunteers also have revealed an association between self-reports of sleep deficit or poor sleep quality and higher Aβ burden in the brain [8,9,10], which is a risk factor for AD. Notice that accumulation of Aβ burden in the brain is associated with impairment of brain functions [11,12]. Thus, effective strategies that prevent Aβ accumulation in the brain can be a promising step in AD therapy and promote healthy brain aging. In this aspect, there is increasing evidence that sleep plays an important role in the protection from Aβ accumulation in the brain [4]. Indeed, in rodents, it has been shown that Aβ clearance from the brain’s ISF predominantly occurs during sleep [6], which is ascribed to the glymphatic pathway, operating most efficiently during sleep [5,13,14]. Clinical studies have also shown that Aβ levels in the cerebrospinal fluid (CSF) are the highest before sleep and the lowest after wakening, while CSF Aβ clearance is conducted by sleep deprivation [15].

The (re)discovery of the meningeal lymphatic vessels (MLVs) has led to a reassessment of the pathways for the clearance of waste from the central nervous system (CNS) [16]. Nowadays, the role of MLVs in brain functions, specifically in the context of ageing and Alzheimer’s disease (AD), has been actively studied [17]. It is interesting to note that Aβ was initially isolated from homogenates of the meningeal tissue from patients with AD [18]. Iliff et al. demonstrated that fluorescent-tagged Aβ (FAβ) is transported along the glymphatic pathway, which can remove FAβ from the central nervous system (CNS) [14]. Mesquita et al. discovered Aβ removal via the MLVs and discussed that the development of methods for modulating the glymphatic/lymphatic clearance of Aβ from the brain could become a revolutionary step in the therapy of AD [17]. Currently, there are no pharmacological drugs that provide an effective AD therapy and limit the development of cognitive impairment [18]. Note that pharmaceutical companies such as Biogen, Johnson & Johnson, and Pfizer announced the cancellation of funding for the synthesis of antibodies for the treatment of AD due to the failure of clinical trials [19]. Obviously, in the next couple of decades, the main strategies for AD treatment will be noninvasive methods for the stimulation of the clearance of toxic Aβ from brain tissues.

Transcranial infrared photostimulation (PS, 1267 nm) is considered a possible novel nonpharmacological and noninvasive therapeutic strategy for the prevention or delay of AD [20,21]. A number of reviews have focused on the application of PS for the treatment of AD [22,23,24,25]. In the last years, there has been accumulating evidence suggesting that PS can reduce Aβ-mediated hippocampal neurodegeneration and memory impairments in rodents, inhibit Aβ-induced brain cell apoptosis, and cause reduction in Aβ plaques in the cerebral cortex [26,27,28,29]. In our previous animal experiments, we clearly showed that PS effectively stimulates the clearing function of the meningeal lymphatic vessels (MLVs), which play a crucial role in Aβ clearance [30,31,32,33]. We also demonstrated that the course of PS during 9 days effectively reduces the Aβ level in mice with AD [33,34]. There is evidence that PS modulates the tone and permeability of the lymphatic vessels, probably through PS-induced stimulation of the synthesis of nitric oxide (NO) in the lymphatic endothelium [26,27,28,29,35]. Thus, PS is a promising candidate as an effective tool for the stimulation of the lymphatic clearance of Aβ from the brain.

In this study, we investigated the nighttime and daytime PS effects on the lymphatic clearance of Aβ from the mouse brain. We clearly demonstrated on healthy mice that the evacuation of Aβ from the right lateral ventricle to the deep cervical lymph nodes (dcLNs) occurs faster at night than during wakefulness. The night course of PS during 9 days improves recognition memory and reduces Aβ accumulation in the brain of AD mice more effectively than the daytime PS course. These pilot results suggest that night PS can be a more effective method in preventing AD and Aβ deposition in the brain than daytime PS.

## 2. Materials and Methods

### 2.1. Subject

Experiments were performed in male BALB/c mice (25 g, total number = 56) in accordance with the Guide for the Care and Use of Laboratory Animals published by the US National Institutes of Health (NIH Publication No. 85–23, revised 1996). The protocols were approved by the Institutional Review Board of Saratov State University (Protocol 5, 29.06.2021). The mice were housed at 25 ± 2 °C, 55% humidity, and 12:12 h light–dark cycle (light: 20:00–08:00). Food and water were given ad libitum. The mice were obtained from a vivarium in Pushchino (Russia) 1 week before the beginning of the experiments to ensure acclimation to the housing room of the animal facility. The experiments were performed in the following groups: (1) awake mice, no PS; (2) awake mice +PS; (3) sleeping mice, no PS; (4) sleeping mice +PS, *n* = 7 in each group in all sessions of the experiments.

### 2.2. An Animal Model of Alzheimer’s Disease in Mice

To induce AD in mice, we used the injection of the Aβ (1–42) peptide (1 μL, 200 μM) in the hippocampus. Aβ (1–42) was dissolved in phosphate-buffered saline (PBS) and then incubated for 5–7 days at 37 °C to induce fibril formation [36]. The mice were anaesthetized by 1% isoflurane at 1 L/min N_2_O/O_2_ (70:30) and fixed in a stereotactic frame. The scalp was removed, and the skull surface was dried by clean compressed air. The injection of Aβ was performed with a 5 µL Hamilton syringe with a 29-G needle at a rate of 0.1 μL/min (Bonaduz, Switzerland) at the coordinates (AP, 2.0 mm; ML, +/–1.3 mm; DV, 1.9 mm).

### 2.3. Monitoring of FAβ Distribution in the Mouse Brain

Ten days before experiments, a polyethylene catheter (PE-10, 0.28 mm ID x 0.61 mm OD, Scientific Commodities Inc., Lake Havasu City, AZ, USA) was implanted into the right lateral ventricle (AP, 1.0 mm; ML, −1.4 mm; DV, 3.5 mm) according to the protocol reported by Devos et al. [37]. A small cranial burr hole was drilled through the skull using a variable speed dental drill (with a 1 mm drill bit). An amount of 5 μL of amyloid-β (1–42), HiLyte™ Fluor 488 labeled (AnaSpec Inc., Fremont, CA, USA), at a rate of 0.1 μL/min was injected into the right lateral ventricle at 08:00 in the group of awake mice and at 20:00 in the group of sleeping mice.

The ex vivo optical study of FAβ distribution was performed 3 h after the intraventricular injection of FAβ in mice, at 11:00 in the groups of awake mice and at 23:00 in the groups of sleeping mice. The imaging was performed using a homemade optical instrument based on the monochrome camera acA2040–2090 um (Basler, Ahrensburg, Germany) and a 50 mm 2.8 C-mount CCTV objective lens (Tamron, Saitama, Japan). The lens was attached to the camera with a 15 mm extension tube to ensure macro imaging with a 23.3 to 31.8 mm field of view depending of the lens focusing ring adjustment. The lens was mounted on the vertical manual translation stage (Standa, Vilnius, Lithuania) above a Petri dish, where samples were submerged in a buffer solution. The top surface of each sample was covered with a 25 mm× 50 mm× 0.17 mm cover glass. The slider with filter sets (49,019, 49,002, Chroma Technology, VT, USA) was placed just below the objective lens. Each filter set was illuminated with homemade condensers with 1W LEDs (635 nm for 49,019 and 460 nm for 49,002) to ensure uniform illumination over the camera field of view. Led illuminators were synchronized with the camera “fire” output.

The camera resolution was 2048 × 2048 pixels at 12 bit grayscale. Images were acquired in a dark room at a constant exposure time of 200 ms, and other settings were kept unchanged for all samples. Image acquisition and processing were performed with a custom software developed using the NI Vision and LabVIEW software (National Instruments, Austin, TX, USA) and the Fiji open-source image processing package [38]. Image processing procedures were identical for each pair of images (control and laser-treated samples) for each channel to ensure an accurate comparison of the fluorescence intensity.

The analysis of FAβ in the brain slices was carried out on a fluorescence microscopic system described above. For a quantitative analysis of the intensity signal from FAβ, ImageJ was used for image data processing and analysis. The intensity of fluorescence for each slide was integrated over a rectangular region of interest bounding the brain slice. The integral value was divided by slice area. The areas of brain slices were calculated using the plugin “Analyze Particles” in the “Analyze” tab, which calculates the total area of FAβ fluorescence intensity tissue elements—the indicator “Total Area”. In all cases, 10 regions of interest were analyzed.

### 2.4. Photomodulation of Aβ Clearance from the Mouse Brain

A fiber Bragg grating wavelength-locked high-power laser diode (LD-1267-FBG-350, Innolume, Dortmund, Germany) emitting at 1267 nm was used as a source of irradiation. The laser diode was pigtailed with a single-mode distal fiber ended by collimation optics to provide a 5 mm beam diameter at the specimen. The nonanesthetized mice with shaved head were fixed using the adapted protocol for two-photonic imaging of the cortical vessels in awake behavior rodents [39] and irradiated in the area of the basal MLVs [40] using a single laser dose of 9 J/cm^2^ (on the skull) and 3 J/cm^2^ on the brain surface or the PS course at 81 J/cm^2^ during 9 days with the sequence of 17 min irradiation and 5 min pause (61 min in total). For the PS course, the mice were treated daily by PS for 9 days, 3 days after the surgery procedure of Aβ injection into the right lateral ventricle. The heating of the brain tissue caused by exposure to light was monitored by using a thermocouple data logger (Pico Technology, USB TC-08, Cambridgeshire, United Kingdom).

### 2.5. Electroencephalography (EEG)

A two-channel cortical EEG/one-channel electromyogram (Pinnacle Technology, Taiwan) was recorded. The mice were implanted with two silver electrodes (tip diameter: 2–3 µm) located at a depth of 150 µm in coordinates (L: 2.0 mm and P: 2 mm) from the bregma on either side of the midline under inhalation anesthesia with 1% isoflurane at 1 L/min N_2_O/O_2_ (70:30). The head plate was mounted, and small burr holes were drilled. Afterward, EEG wire leads were inserted into the burr holes on one side of the midline between the skull and the underlying dura. EEG leads were secured with dental acrylic. An EMG lead was inserted in the neck muscle. Ibuprofen (15 mg/kg) for the relief of postoperative pain was provided in their water supply for 2 to 3 days prior to surgery and for 3 days postsurgery. The mice were allowed 10 days to recover from surgery prior to beginning the experiment.

Since the standard sleep staging rules for rodents are currently not available, we referred to the visual scoring criteria from [41,42,43,44]. Wakefulness, nonrapid eye movement (NREM), and rapid eye movement (REM) sleep were visually scored in 10 s epochs. EEG activity was measured and compared in mice in awake state and during sleep. Wakefulness was defined as a desynchronized EEG with low-amplitude and high-frequency dynamics (>10%, 8–12 Hz) and relatively high-amplitude electromyography (EMG). NREM sleep was recognized as synchronized activity with high amplitude, which is dominated by low-frequency delta waves (0–4 Hz) comprising >30% of EEG waveforms/epoch and a lower-amplitude EMG. REM was identified by the presence of theta waves (5–10 Hz) comprising > 20% of EEG waveforms/epoch with a low EMG amplitude.

### 2.6. Object Recognition Test (ORT) 

The ORT is the most common behavior test for mice with AD, which is used for the evaluation of recognition memory [45,46]. The mice were presented with two similar objects (blue cubes) during the first session (familiarization session), and then one of the two cubes was replaced by a new object (pink ball) during the second session (test session). Because it has been demonstrated that prior experience can alter the behavioral responses of mice in the ORT [47], animals were accustomed to being handled by experimenters twice a week for 1 min each session for 1 week before the beginning of the experiments. Following the protocol of Sik et al. [48], the habituation phase consisted of 5 min exposures to the testing arena per day separated by 6 h during 3 days before the test phase. The test was conducted 10 days after surgery. Two identical cubes were placed in a cage for mice for 10 min. Then there was a second 10 min session when one cube was replaced with an unfamiliar earlier ball [45]. The exploration time for both objects during the test phase was 20 s [45]. For the experiments, we chose to use a black wooden box (33 cm × 33 cm × 20 cm) using a video-tracking package. We used two asymmetric objects of the same size (cubes: 3 cm × 3 cm × 3 cm; ball: 3 cm in diameter) and odor. Since mice have difficulty in discriminating colors, we selected bright (dark blue and pink) objects. The weights of the objects were heavy enough that the mice could not move them. Additionally, we used Patafix to hold the objects stuck on the floor. The mice were housed in a light–dark cycle and were tested in the dark phase (active phase between 08:00 and 20:00). The familiarization session was carried out in the morning (at 09:00). The mice were placed in the testing room 30 min before testing. The exploration was defined as follows: directing the nose toward the object at a distance of less than or equal to 2 cm. We chose to score the object exploration whenever the mouse sniffed the object or touched the object while looking at it (i.e., when the distance between the nose and the object was less than 2 cm). Climbing onto the object (unless the mouse sniffs the object it has climbed on) or chewing the object does not qualify as exploration.

### 2.7. Immunohistochemical (IHC) Assay

The mice were euthanized with an intraperitoneal injection of a lethal dose of ketamine and xylazine. Afterward, the mice were decapitated; their dcLNs were removed and fixed in 4% buffered paraformaldehyde for 2 days and in 20% sucrose for another day. To label the lymphatic vessels, samples were incubated overnight at +4 °C with goat anti-rabbit Lyve1 and Prox1 antibody (1:500; Invitrogen, Molecular Probes, Eugene, OR, USA). After several rinses in PBS, the samples were incubated for 3 h at room temperature with fluorescent-labeled secondary antibodies on 1% BSA/0.2% Triton X-100/PBS (1:500; goat anti-rabbit IgG (H + L) Alexa Fluor 405 and goat anti-mouse IgG (H + L) Alexa Fluor 647; Invitrogen, Molecular Probes, Eugene, Oregon, OR, USA) with further confocal analysis (Nikon Corp., Tokyo, Japan).

For histological analysis of Aβ depositions in the cortex, we used the protocol for IHC analysis with anti-Aβ antibody (1:500; Abcam, ab182136, Cambridge, UK). The tissue samples were fixed with formaldehyde and, after a routine processing, were embedded into a paraffin block. Then the samples were sectioned into 3 to 5 μm sections on a vibratome (Leica VT1000 S Microsystem); the sections were attached to poly-L-lysine-coated glass slides; they were dried at 37 °C for 24 h; and they were sequentially incubated in xylene (three times, 3 min each), 96% ethanol (three times, 3 min each), 80% ethanol (two times, 3 min each), and distilled water (three times, 3 min each). The IHC reaction was visualized with a REVEAL—BiotinFree Polyvalent diaminobenzidine kit (Spring Bioscience, Pleasanton, CA, USA). Endogenous peroxidases were blocked by adding 0.3% hydrogen peroxide to the sections for 10 min, followed by washing of sections in PBS. The antigen retrieval was conducted using a microwave oven in an ethylenediaminetetraacetic acid buffer pH 9.0, and a nonspecific background staining was blocked in PBS containing 0.5% bovine serum albumin and 0.5% casein for 10 min, after which the sections were washed in PBS for 5 min. Further, the sections were incubated in a humid chamber with diluted anti-Aβ antibody (1:500; Abcam, ab182136, Cambridge, USA) for 1 h at room temperature. After that, the sections were washed in PBS, incubated with secondary horseradish peroxidase-labeled goat anti-rabbit antibodies for 15 min, again washed in PBS, counterstained with hematoxylin for 1 min, washed again in water, dehydrated in graded alcohols (three times, 3 min each) and then in xylene (three times, 3 min each), and finally embedded into Canadian balm. The histological sections were evaluated by light microscopy using a Mikrovizor μVizo-103 digital image medical analysis system (LOMO, St. Petersburg, Russia). Approximately 10 slices per animal were imaged.

The quantitative analysis of the results of IHC staining on an Aβ marker was carried out on a microscopic system with automatic analysis of the obtained photos of Ariol SL50 (Genetix, New Milton, Hampshire, UK). The total area of IHC was determined with the software module PathVysion (Music, Inc., Applied Imaging, Tampa, FL, USA).

### 2.8. Statistical Analysis

All statistical analysis was performed using Microsoft Office Excel and SPSS 17.0 for Windows software. The results were reported as a mean value ± standard error of the mean (SEM). The differences in the signal intensity of FAβ in dcLNs and the immunopositive Aβ in the brain were evaluated using the two-sample Welch’s *t*-test and nonparametric Mann–Whitney U tests. The mean time to reach the criterion between the two sessions in ORT was compared with a one-way ANOVA. The significance levels were set at *p* < 0.05 for all analyses. No statistical methods were used to predetermine the sample size.

## 3. Results

### 3.1. EEG Analysis of Sleep Stages and Wakefulness

Since the standard sleep staging rules for mice are not available currently, we developed our original approach based on a Fourier method for the recognition of wakefulness and REM and NREM sleep as well as for the analysis of the spectrum of the EEG signals [42]. Figure 1c shows the allocation of four frequency bands for alpha, beta, theta, and delta waves with their integrated power. The wakefulness, NREM, and REM stages were determined by comparing the power indices of the EEG signals (Figure 1d). Figure 1d illustrates that the transition of mice to sleep, both NREM and REM, is characterized by an increase in the total amplitude of EEG signals. At the same time, the amplitude of EEG waves in the NREM stage significantly exceeds that in the REM stage (Figure 1d).

Additionally, using recommendations for the nonlinear analysis of EEG signals at various sleep stages [43,44], we determined the phase of the NREM stage when the power of the alpha and beta bands was less than for the theta and delta waves. REM sleep was detected when the power of the alpha and beta bands coincided with delta and theta (Figure 1c). Figure 1e reflects the average duration of the waking state, NREM, and REM stages. Using our method of analysis of EEG signals, we determined the delta band for PS of the dorsal part of the MLVs at the other stages of the experiments.

### 3.2. PS-Mediated Stimulation of Lymphatic Clearance of FAβ from the Mouse Brain at Night and During the Day

In this step of the experiments, we studied the PS effects on the lymphatic clearance of FAβ from the brain of the mice in the four groups: (1) awake mice, no PS; (2) awake mice +PS; (3) sleeping mice, no PS; (4) sleeping mice +PS. Figure 2a,d illustrates day (at 08:00 a.m.) and night (08:00 p.m.) injection of FAβ in the right lateral ventricle via an implanted catheter to avoid the anesthesia effects of the lymphatic clearance of FAβ [49]. Our results clearly demonstrate that the intraventricular injection of FAβ was accompanied by the distribution of a tracer in the brain of the mice in all the tested groups. Figure 2e shows that the spread of FAβ was higher on the ventral than on the dorsal aspect of the brain 3 h after its injection. These data reflect the FAβ distribution from the ventricle to the basal MLVs, playing an important role in the brain drainage and clearance [16,17,30,31,32,33]. The dcLNs are the first anatomical station of the CSF exit with dissolved unneeded substances for the brain [16,17,30,31,32,33]. Therefore, the more intensively the substance is released from the brain, the stronger is its level in dcLNs. Figure 2e–g demonstrates the qualitative and quantitative analyses of the lymphatic clearance of FAβ from the brain and its accumulation in dcLNs.

Our results revealed that the signal intensity from FAβ in dcLNs was higher in both the awake and sleeping mice treated with PS than in the untreated groups. However, the group of sleeping mice +PS demonstrated a stronger signal from FAβ in dcLNs vs. the group of awaking mice +PS, suggesting a more effective PS-mediated stimulation of the lymphatic clearance of FAβ at night than during the day. Notice that sleep was accompanied by a higher accumulation of FAβ in dcLN than wakefulness, suggesting a natural activation of FAβ clearance from the brain [5], which was significantly increased after PS (Figure 2g). Thus, this series of experiments demonstrates that PS stimulates the clearance of FAβ from the mouse brain more effectively at night than during the day, which can be due to natural night activation of Aβ removing from the brain.

### 3.3. Effects of Night and Daytime PS Course on Aβ Deposition in the Brain and on Recognition Memory in AD Mice

In the final step, we analyzed the effects of the night and day courses of PS during 9 days on the behavior of the mice using the ORT and on the deposition of Aβ in the brain of the AD mice. Figure 3a illustrates the design of the experiment. Ten days after intrahippocampal injection of Aβ, the PS course was conducted, and ORT was performed 1 day after the PS course. Afterward, the brain samples were taken for the IHC analysis. Figure 3b,c shows that the AD mice demonstrated a high deposition of Aβ in the brain, while the PS course significantly reduced the presence of Aβ in the brain parenchyma. However, the night PS course was accompanied by a stronger reducing deposition of Aβ in the brain than the day PS course. The ORT revealed a recognition memory deficit in the AD mice without PS (Figure 3d). Indeed, they demonstrated the same time for exploring the familiar and new objects, which indicates memory impairment. The daytime and nighttime PS courses improved the memory status of the AD mice. Therefore, in both groups, the AD mice treated with PS spent more time for the exploration of the new subject than the familiar one. However, the AD mice, after the night PS course, performed the ORT much faster than the mice after the day PS course (Figure 3d). Thus, the PS course at night was accompanied by a stronger clearance of Aβ from the brain of the AD mice, which was associated with a more effective improvement of recognition memory than the PS course during the day.

## 4. Discussion

Here, we investigated the effects of infrared PS (1267 nm) on the lymphatic removal of Aβ from healthy and AD mouse brains at night and during the day. An infrared light of 800–1100 nm was widely used for the PS therapy of brain diseases, including AD [22,50]. However, infrared PS has a significant limitation, such as limited penetration into the brain due to light scattering and heating effect [51]. A light wavelength of 1300 nm has less scattering and can penetrate deeper into the brain [52]. Therefore, we selected PS (1267 nm, 9 J/cm^2^) for the modulation of the lymphatic clearance of Aβ from the brain. The choice of a PS dose of 9 J/cm^2^ was based on our recent results demonstrating that PS at 9 J/cm^2^ is an optimal dose for the stimulation of the lymphatic clearance of different tracers and red blood cells from the right lateral ventricle without the heating effect [35].

Our results clearly demonstrate that PS at night is accompanied by a faster FAβ clearance from the ventricle to dcLNs in healthy mice as well as a more effective improvement of recognition memory and a reduction of the Aβ deposition in the brain of AD mice than PS during the day. We assume that it is due to natural night activation of the Aβ clearance from the brain. Therefore, sleep is associated with a 60% increase in the interstitial space, resulting in a striking increase in the connective exchange of CSF with ISF [5]. In turn, convective fluxes of ISF increase the rate of Aβ clearance during sleep. Indeed, we found that FAβ clearance was faster in sleeping mice vs. awake animals, which confirms the idea of night Aβ clearance. However, our results do not match Xie’s data [5], suggesting a significant suppression of Aβ clearance in awake mice. We observed Aβ clearance also in wakefulness, which was much lesser than in sleeping animals. Sleep is accompanied by a higher accumulation of FAβ in dcLN than wakefulness, suggesting a natural activation of FAβ clearance from the brain [5], which was significantly increased after PS. We believe that night PS has synergic effects on the natural activation of mechanisms underlying the night lymphatic drainage of the brain parenchyma, which explains the more effective PS influences on Aβ clearance from the brain at night than during the day.

The possible explanation for the more effective night vs. daytime PS stimulation of the lymphatic clearance of Aβ from the brain can be circadian oscillations of the Aβ level in the brain. It has been shown that Aβ clearance from the mouse brain predominantly occurs during sleep [6]. In humans, CSF Aβ levels are highest before sleep and lowest after wakening [15]. The metabolisms and drainage of the brain tissues have a circadian clock, which is regulated by many factors, including hormones. Arousal is driven by the concerted release of neuromodulators [53]. In particular, locus coeruleus-derived noradrenergic (NA) signaling appears critical for driving cortical networks into the awake state of processing [54,55]. The anterior and lateral hypothalamus [56], cervical spinal cord [57], midbrain [58], caudate nucleus [59], medial lower brain stem [60], and frontal cortex [61] all have significant circadian variations in their NA content. The NA suppresses the drainage and clearance of the brain tissues, which is associated with a reduced CSF tracer influx during the day [55]. An interesting note is that the administration of NA antagonists into the cisterna magna causes an increase in the CSF tracer influx as during sleep [5].

In our previous works, we studied the mechanisms of the PS effects on the lymphatics. Our data demonstrate that PS induces relaxation of the lymphatic vessels, which is associated with an increase in the permeability of the lymphatic endothelium [30,31,32,35]. We show that a decrease in the transepithelial electrical resistance and in the expression of tight junction (TJ) proteins, such as CLDN-5, VE-cadherin, and ZO-1, might be a possible mechanism underlying a PS-mediated increase in the lymphatic permeability. TJ proteins are structural compounds of mature lymphatic vessels and play an important role in moving the interstitial fluid and immune cells through the lymphatic endothelium [62]. The lymphatic permeability is actively regulated by several signaling pathways, among which nitric oxide (NO) has been studied in more detail [63]. NO is a vasodilator that acts via the stimulation of soluble guanylate cyclase to form cyclic GMP (cGMP), which activates the protein kinase G, causing the opening of calcium-activated potassium channels and the reuptake of Ca^2+^. The decrease in the concentration of Ca^2+^ prevents myosin light-chain kinase from phosphorylating the myosin molecule, leading to the relaxation of lymphatic vessels [64]. There are several other mechanisms by which NO could control the lymphatic tone and contractility: (1) the activation of an iron regulatory factor in macrophages [65], (2) the modulation of proteins such as ribonucleotide reductase [66] and aconitase [67], the stimulation of the ADP-ribosylation of glyceraldehyde-3-phosphate dehydrogenase [68] and protein-sulfhydryl-group nitrosylation [69].

PS increases the lymphatic contractility [30,35] that is regulated by NO [62,63]. Our recent findings show that PS dilates the MLVs and increases NO production in the isolated lymphatic endothelial cells, which is associated with an increase in the lymphangion contraction [35]. We assume that the PS-mediated activation of the NO synthesis in the lymphatic endothelium leads to the activation of the lymphatic contractility, which might be the possible mechanisms responsible for the PS stimulation of the lymphatic clearance of cells and macromolecules from the brain. The described effects can be related to a PS-mediated increase in the activity of endothelial NO synthase [70].

We hypothesize that the PS effects on the tone and permeability of the MLVs can facilitate the drainage of the brain tissues, which drains Aβ from the ventricles to the subarachnoid space, where Aβ partly penetrates in the MLVs with further Aβ removal in dcLNs.

## 5. Conclusions

In sum, our results clearly demonstrate that PS at night is accompanied by a faster lymphatic clearance of Aβ from the brain in healthy mice as well as a more effective improvement of recognition memory and a reduction of Aβ deposition in the brain of AD mice compared with PS during the day. These data suggest that night PS is a more promising candidate than daytime PS for preventing Aβ accumulation in the brain and AD therapy.

## Figures and Tables

**Figure 1 cells-10-03289-f001:**
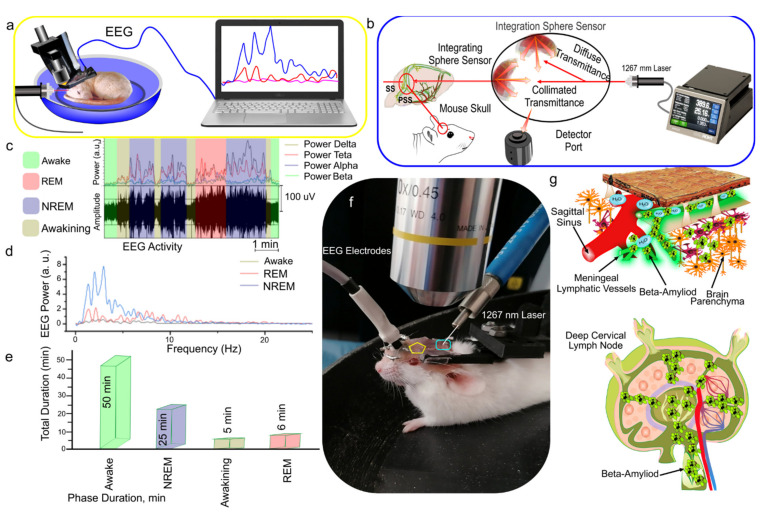
Schematic illustration of the design of the experiments: (**a**) EEG recording in the mouse; (**b**) schema of the laser effects on the dorsal part of the MLVs [40]; (**c**) changes in the power of the EEG signal in the delta, theta, alpha, and beta ranges; (**d**) spectrum of the EEG signal in three different states: wakefulness, REM, and NREM sleep; (**e**) average duration of wakefulness, REM, and NREM sleep; (**f**) representative image of simultaneous EEG recording and PS effects on the dorsal part of the MLVs of nonanesthetized mouse; (**g**) schema of the PS-mediated stimulation of the lymphatic clearance of FAβ from the mouse brain via the MLVs into dcLN.

**Figure 2 cells-10-03289-f002:**
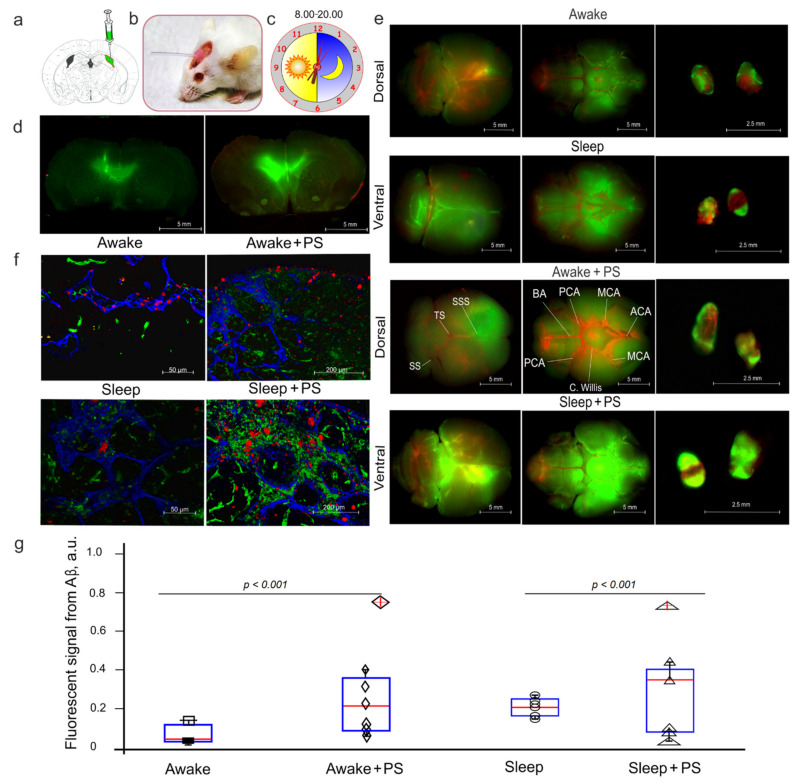
Photostimulation of the lymphatic clearance of FAβ in sleeping and awake mice: (**a**) schema of the intraventricular injection of Faβ, (**b**) representative image of a mouse with an implanted catheter in the right later ventricle, (**c**) The time of the intraventricular injection of Faβ, (**d**) brain sections of the intraventricular injection of Faβ, (**e**) representative images of the distribution of FAβ in the dorsal and ventral aspects of the brain and the accumulation of FAβ in dcLNs 3 h after the intraventricular injection of FAβ in the four tested groups, (**f**) representative images of the presence of FAβ in the Lyve1/Prox1 lymphatic vessels of dcLNs 3 h after the intraventricular injection of FAβ in the four tested groups, (**g**) quantitation of the signal intensity of FAβ in the four tested groups (the two-sample Welch’ *t*-test and nonparametric Mann–Whitney U tests, *n* = 7 in each group).

**Figure 3 cells-10-03289-f003:**
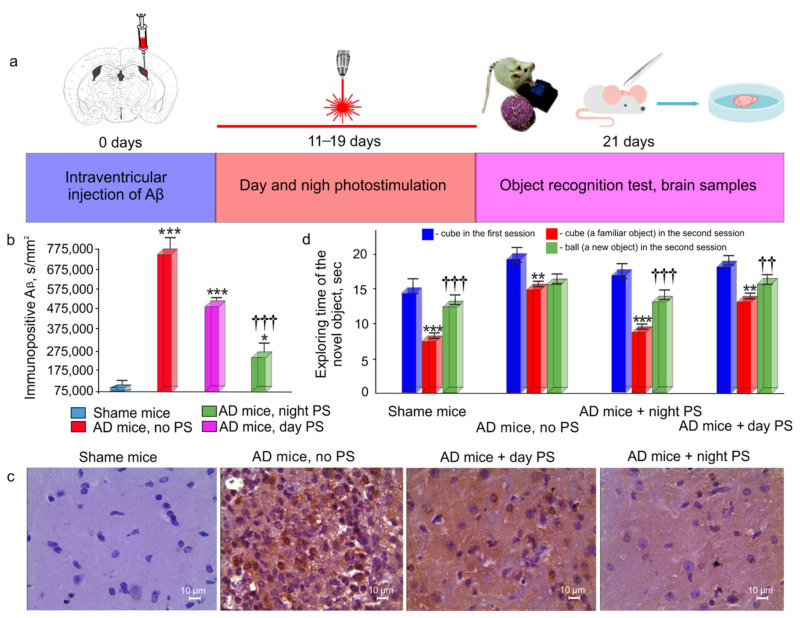
Effects of the 9-day course of PS on the Aβ deposition in the brain and on recognition memory in the AD mice: (**a**) time points for the intrahippocampal injection of Aβ, PS application, ORT, and the brain collection; (**b**) quantitation of Aβ in the brain of the mice in the four tested groups (*** - *p* < 0.001, * < 0.01 vs. the sham group, ^†††^ - *p* < 0.001 between the groups treated by PS, two-sample Welch’s *t*-test, and nonparametric Mann–Whitney U tests, *n* = 7 in each group); (**c**) representative IHC images of Aβ in the brain of the mice in the four tested groups; (**d**) object recognition test, reflecting recognition memory (*** < 0.001, ** < 0.05 vs. cubes in the first session; ^†††^ - *p* < 0.001, ^††^ - *p* < 0.05 vs. ball in the second session, a one-way ANOVA, *n* = 7 in each group).

## Data Availability

The data that support the findings of this study are available on request from the corresponding author.

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
