# Peer review of "Night Photostimulation of Clearance of Beta-Amyloid from Mouse Brain: New Strategies in Preventing Alzheimer’s Disease"

_cells, 2021, doi:10.3390/cells10123289_

Round 1

Reviewer 1 Report

The manuscript faces a very challenging and novel problem suggesting a potential application of brain photostimulation ( at night) preventing Abeta deposition in AD patients.  The experimental design is quite complete and detailed, and its explanation very clear and effective. 

may add something again the fact that many cerebral functions exhibit some circadian rhythm ( e.g. adenosine accumulation for sleep induction) and therefore also Abeta deposition during the day and clearance during the night is sensible. 

Some details missing in the intro (e.g. laser characteristics,...) are actually explained in the discussion.

Also the final explanations are solidly based. There are a few typos to be corrected: line 24 (stimulation) ,34 (delete comma), 38 (present ), 59 (stimulation),95 ( space btw by and clean),100 (into) ,110 ( attached on the camera) ,372 (recent).

some other misspellings: fig. 3 'night' ; fig.1  last picture 'beta-amyloid'.

Author Response

Comment: The manuscript faces a very challenging and novel problem suggesting a potential application of brain photostimulation ( at night) preventing Abeta deposition in AD patients.  The experimental design is quite complete and detailed, and its explanation very clear and effective.

May add something again the fact that many cerebral functions exhibit some circadian rhythm ( e.g. adenosine accumulation for sleep induction) and therefore also Abeta deposition during the day and clearance during the night is sensible.

Response: We would like to express our deep appreciation for the great support from the reviewer and for the constructive advices. We added in Discussion the possible explanation of more effective night vs. daytime PS stimulation of lymphatic clearance of Aβ from the brain due to circadian oscillations of Aβ level in the brain. We also discuss the circadian rhythm of norepinephrine, which plays an important role in the regulation of day-night changes in drainage and clearance of the brain tissues. All changes are highlighted in yellow.

Comment: Some details missing in the intro (e.g. laser characteristics,...) are actually explained in the discussion.

Response: We added in Introduction the laser characteristics and more details about the mechanisms of photomodulation of lymphatic functions.

Comment: Also the final explanations are solidly based. There are a few typos to be corrected: line 24 (stimulation) ,34 (delete comma), 38 (present ), 59 (stimulation),95 ( space btw by and clean),100 (into) ,110 ( attached on the camera), 372 (recent). Some other misspellings: fig. 3 'night' ; fig.1  last picture 'beta-amyloid'.

Response: All grammatical and stylistic errors, Figures 1 and 3 as well as the English language, have been corrected.

We thank the reviewer again for the positive support and highly qualified assessment of our results. This is an important experience for us in the preparation of our article in Cells.

Authors

Reviewer 2 Report

Semyachkina-Glushkovskaya et al. present here their work on modulating Aβ clearance through photostimulation (PS) and propose it as a valuable strategy for AD prevention. Particularly, they demonstrate that PS performed at night clears Aβ deposits in mice brains. The study and related results are interesting, but the manuscript has too many flaws, especially in the Discussion and Conclusions sections. 

  • Authors partially build discussion around previous results collected by them. The cited article is not online published on Nature Communications as listed by authors. Please provide doi of this manuscript. Otherwise, authors should provide more evidence to support their discussion.
  • Conclusions are repetitive and not realistic. 
  • In the introduction, authors should better explain and introduce photo-stimulation.
  • In Material and Methods fluorescent Abeta is abbreviated as FAβ. Then, suddenly authors refers to it as FAα or Fαβ, to randomly switch back to FAβ. Please explain. 
  • Throughout the test there are too many mistakes and typos such "attaché o the cam-era" (110). Please check the text and correct all of them. I would also suggest an extensive revision by an English expert.
  • Figure 3d is not clear, neither is the related caption.

Author Response

Comments: Semyachkina-Glushkovskaya et al. present here their work on modulating Aβ clearance through photostimulation (PS) and propose it as a valuable strategy for AD prevention. Particularly, they demonstrate that PS performed at night clears Aβ deposits in mice brains. The study and related results are interesting, but the manuscript has too many flaws, especially in the Discussion and Conclusions sections. Authors partially build discussion around previous results collected by them. The cited article is not online published on Nature Communications as listed by authors. Please provide doi of this manuscript. Otherwise, authors should provide more evidence to support their discussion. Conclusions are repetitive and not realistic. In the introduction, authors should better explain and introduce photo-stimulation.

Response: We would like to express our gratitude to the reviewer for the valuable advices and recommendations. We improved Introduction, Discussion and Conclusion. We corrected the error in the references list and added doi of our manuscript, which is under review in Nat Com (second round) and is submitted in Biorxiv. All changes are highlighted in yellow.

Comments: In Material and Methods fluorescent Abeta is abbreviated as FAβ. Then, suddenly authors refers to it as FAα or Fαβ, to randomly switch back to FAβ. Please explain. Throughout the test there are too many mistakes and typos such "attaché o the camera" (110). Please check the text and correct all of them. I would also suggest an extensive revision by an English expert. Figure 3d is not clear, neither is the related caption.

Response: All grammatical and stylistic errors, Figure 3 as well as the English language, have been corrected. In the text of manuscript should be FAβ (fluorescent amyloid-β), we made correction.

We would like to thank the reviewer for the great work in reviewing our article and useful recommendations for improving of our paper in Cells.

Authors

Round 2

Reviewer 2 Report

I would like to thank the authors to have properly answered my comments.

There are still just a couple of few minor mistakes to be corrected:

- line 129: Correct "in according to the protocol" with "according to the protocol reported by Devos et al."
- in section 3.2 and related Figures/captions, authors report FAα instead of FAβ several times. Please correct.

line 463: "We hypothesis" should be corrected with "We hypothesize" or "Our hypothesis is" 

Author Response

Commens: There are still just a couple of few minor mistakes to be corrected:

- line 129: Correct "in according to the protocol" with "according to the protocol reported by Devos et al."
- in section 3.2 and related Figures/captions, authors report FAα instead of FAβ several times. Please correct.

line 463: "We hypothesis" should be corrected with "We hypothesize" or "Our hypothesis is" 

Response: We would like to thank the reviewer again for the great support of our research. It is a great honor for us to gain invaluable experience with the help of the reviewer in improving our article for publication in Cells. 

We made corrections of typos and phrases. In the whole text is FAβ now.  

Authors